# R$^2$SFD : Improving Single Image Reflection Removal using Semantic Feature Dictionary

### Green Rosh*
Samsung R&D Institute India, Bangalore
Bengaluru, India
greenroshks@gmail.com

### Pawan Prasad B H*
Samsung R&D Institute India, Bangalore
Bengaluru, India
pawan.prasad@samsung.com

### Lokesh R Boregowda
Samsung R&D Institute India, Bangalore
Bengaluru, India
lokesh.rb@samsung.com

### Kaushik Mitra
Indian Institute of Technology, Madras
Chennai, India
kmitra@ee.iitm.ac.in

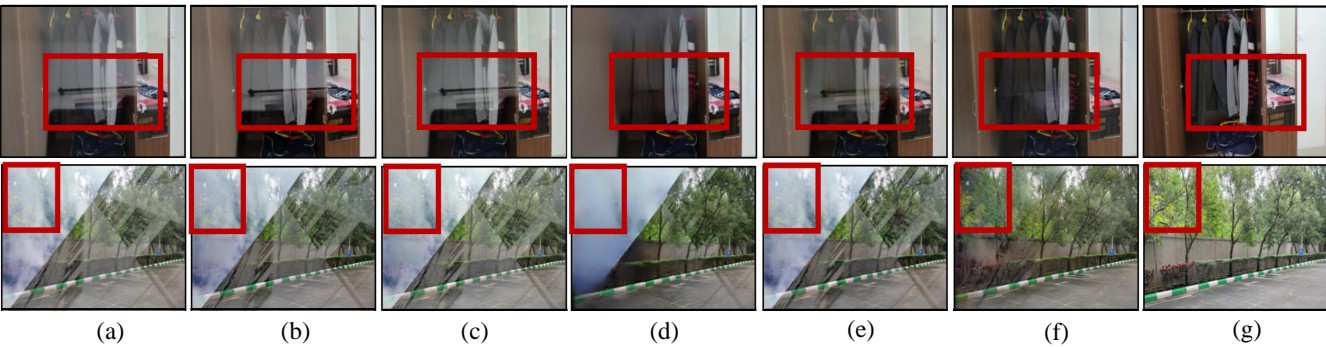

|       |       |       |       |       |       |       |
|-------|-------|-------|-------|-------|-------|-------|
| (a)   | (b)   | (c)   | (d)   | (e)   | (f)   | (g)   |

Figure 1: Improving Reflection Removal using Semantic Feature Dictionary. (a) Input (b) ERRNet [51] (c) IBCLN [21] (d) Dong *et.al* [4] (e) RAGNet [25] (f) Ours (g) Ground Truth. Our method (f) is able to remove reflections much better than state-of-the-art methods in challenging scenarios (Red boxes).

## Abstract

Single image reflection removal is a severely ill-posed problem and it is very hard to separate the desirable transmission and undesirable reflection layers. Most of the existing single image reflection removal methods try to recover the transmission layer by exploiting cues that are extracted only from the given input image. However, there is abundant unutilized information in the form of millions of reflection free images available publicly. Even though this information is easily available, utilizing the same for effectively removing reflections is non-trivial. In this paper, we propose a novel method, termed *R$^2$SFD*, for improving single image reflection removal using a Semantic Feature Dictionary (SFD) constructed from a database of reflection-free images. The SFD is constructed using a novel *Reflection Aware* Feature Extractor (RAFENet) that extracts features invariant to the presence of reflections. The SFD and the input image are then passed to another novel network termed SFDNet. This network first extracts RAFENet features from the reflection-corrupted input image, searches for similar features in the SFD, and transfers the semantic content to generate the final output. To further improve reflection removal, we also introduce a Large Scale Reflection Removal (LSRR) dataset consisting of 2650 image pairs comprising of a variety of real world reflection scenarios. The proposed method achieves superior results both qualitatively and quantitatively compared to the state of the art single image reflection removal methods on real public datasets as well as our LSRR dataset. We will release the dataset at https://github.com/ee19d005/r2sfd.

## CCS Concepts

• **Computing methodologies → Computational photography**.

## Keywords

Reflection Removal; Deep Learning; Semantic Search

**ACM Reference Format:**
Green Rosh, Pawan Prasad B H, Lokesh R Boregowda, and Kaushik Mitra. 2024. **R$^2$SFD** : Improving Single Image Reflection Removal using Semantic Feature Dictionary. In *Proceedings of the 32nd ACM International Conference on Multimedia (MM '24), October 28-November 1, 2024, Melbourne, VIC, Australia*. ACM, New York, NY, USA, 10 pages. https://doi.org/10.1145/3664647.3681450

---

*Both authors contributed equally

## 1  Introduction

Capturing images in the presence of obstructions such as glass typically results in unwanted reflections. An image $\mathbf{I} \in \mathbb{R}^{m \times n}$ corrupted by reflections can be modeled as a combination of two layers - the desirable transmission layer $\mathbf{B} \in \mathbb{R}^{m \times n}$ and the undesirable reflection layer $\mathbf{R} \in \mathbb{R}^{m \times n}$ [41]. This can be represented as follows:

$$I = I_t + \alpha.I_r \tag{1}$$

where $I$, $I_t$ and $I_r$ denotes the reflection corrupted image, the transmission layer and the reflection layer respectively. The blending factor $\alpha$ denotes the strength of reflection, which is typically dependent on imaging conditions. The restoration of such an image involves layer separation which is severely ill-posed [54]. There have been many works in the past addressing the problem of reflection removal. Several earlier methods attempt to constrain the solution space using hand-crafted priors such as natural scene statistics [20], ghosting cues [42] and gradient sharpness [54]. However, these hand-crafted priors often cater to specific kinds of reflections and is not generalizable to handle a wider range of in-the-wild reflections. More recently, deep learning based approaches have been observed to produce state-of-the-art results in reflection removal [3, 6, 8, 9, 12, 21, 25, 26, 34, 56, 58]. Even though deep learning methods have made significant strides, single image reflection removal still remains a significantly hard problem due to its ill-posed nature.

The existing deep learning based reflection removal methods have tried to exploit cues that discriminate between transmission and reflection layers. These cues are either learned in a multi-stage pipeline [4, 6, 21, 56] or enforced using sophisticated loss functions [51, 59]. However, these cues are extracted using only the information present in the given input image. The dependence on reflection-corrupted input images hamper the ability of the neural network to effectively recover the transmission layer, especially in regions with high reflections. To alleviate this challenge, we propose to utilize information from external reflection-free images with similar semantic content as that of the input image. We hypothesize that the higher level semantic information from reflection-free images can guide the network to learn a better representation of the underlying scene. An earlier method [50] proposed a non-deep learning based approach using non-local image priors extracted from external images. However, this method uses hand-crafted sparsity priors, which may not generalize well for in-the-wild reflections. To the best of our knowledge, there exists no deep learning based method for the extraction and utilization of reflection-free semantic content from a database of images for improving reflection removal.

In this paper, we propose a novel deep learning based method that can automatically retrieve and utilize reflection-free semantic content from a large image database to effectively remove reflections (see Fig. 2). Given an input image, we first use a light-weight Coarse Semantic Search (CSS) module to select a set of similar images from a large database of reflection-free images. These images are then passed through a novel *Reflection Aware* Feature Extractor (RAFENet) to construct a semantic feature dictionary (SFD), consisting of image features. The RAFENet is trained using a novel Reflection Aware loss function to ensure that the extracted features are mostly invariant to reflections. The SFD along with the

reflection-corrupted input image is then passed to a novel architecture, termed Semantic Feature Dictionary Network (SFDNet), to generate the final output. The proposed SFDNet is designed to utilize the information from the semantic feature dictionary to improve reflection removal from the input image. As shown in Fig. 1, our method is able to remove reflections much better than the state-of-the-art methods, especially in regions with very high reflections.

We also propose a new dataset for advancing the progress in reflection removal. The existing datasets for reflection removal are either small-scale ($\sim$ 1000 image pairs) [17, 49] or consists of only reflection layer information [48]. To the best of our knowledge, our dataset, termed Large Scale Reflection Removal (LSRR) dataset, consisting of 2650 image pairs, is the largest dataset containing real paired data for reflection removal. The major contributions of this paper are as follows:

**(a)** We propose a novel method for improving reflection removal by utilizing reflection-free semantic content from external image databases. To the best of our knowledge, this is the first deep learning based method that utilizes reflection-free image databases for improving single image reflection removal.

**(b)** We also introduce a new dataset termed Large Scale Reflection Removal (LSRR) dataset consisting of 2650 image pairs. To the best of our knowledge, the proposed dataset is the largest paired real dataset for reflection removal.

**(c)** The proposed method outperforms state-of-the-art methods in reflection removal both qualitatively and quantitatively on public real datasets as well as our LSRR dataset

## 2  Related Works

In this section, we provide an overview of the existing literature in reflection removal.

**Single Image Reflection Removal:** Earlier works in reflection removal used hand-crafted priors such as natural image statistics [19, 20], relative smoothness [24], gradient priors [2, 39, 45, 50], ghosting cues [42] [10] and depth-of-field differences [47] to remove reflections from images. However, the hand-crafted priors used in these methods often fail in several real-life situations, resulting in lack of generalizability for these methods. To address this issue, several deep learning based methods with novel network enhancements have been proposed in the recent past, which generate state-of-the-art results [3, 6, 8, 9, 12, 21, 25, 26, 34, 56, 58]. Several other methods proposed novel loss functions for reflection removal such as adversarial loss [59], alignment invariant loss [51] and gradient loss [48]. Another method proposed a self-semantic segmentation guided approach using a multi-task network [26]. There have also been several works which proposed novel methods for synthetic data modeling such as [13, 52, 61]. There has also been methods exploring methods that are not fully supervised such as [14, 29, 37].

**Multi Image Reflection Removal:** Earlier works in multi-image reflection removal explored approaches using gradient sparsity interdependence [7], SIFT flow [23] and video based methods [31, 43]. Lun *et.al* [27] proposed a deep learning based approach to remove reflections using optical-flow based alignment, while Prasad *et.al* [33] proposed a deep learning based approach to remove reflections

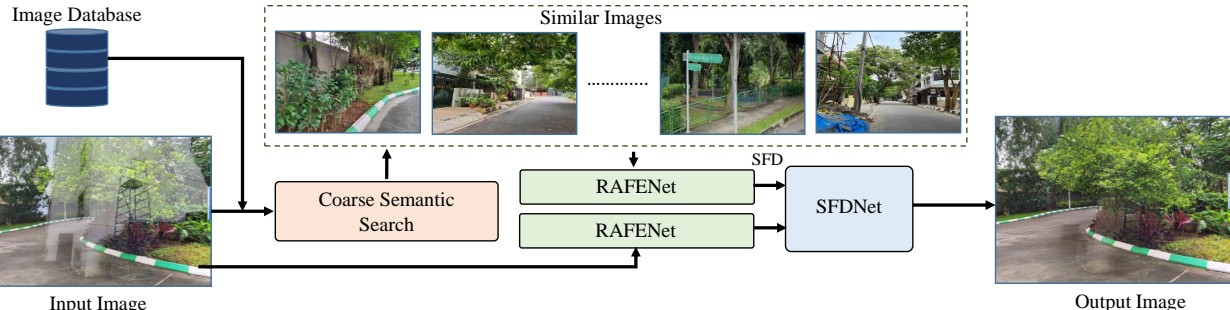

**Figure 2: Overview of the proposed method. A coarse semantic search is first performed on a database of images to find images similar to input. These images are then passed through the RAFENet to generate a Semantic Feature Dictionary (SFD). The input image and the SFD is then passed to SFDNet to generate the reflection-free output.**

from a burst of images. A method for reflection removal using a reference image provided by the user was proposed by [35]. Several other methods focused on reflection removal using images captured using multiple modalities such as multi-focal lengths [40], dual-pixel sensors [36] and different flash or exposure settings [16] [1]. Several other works such as [18] [22] [53] used polarized images to extract cues that helps in reflection removal.

**Reference Based Image Restoration:** Reference based image restoration using deep learning have been studied in domains such as image super-resolution [11, 28, 32, 55, 60] and image deblurring [15, 38]. These methods try to match the features extracted from the input and reference images to find relevant semantic content. These methods mostly use pre-trained VGG features for this purpose [44]. Further, most of these methods require the user to explicitly provide a reference image along with the input image. Our method is different from these approaches in two ways. First, our method does not require any manual intervention for choosing any reference image and is fully automatic during inference. Second, our method uses a novel feature extraction module to extract *reflection-aware* features that are invariant to reflections unlike the reference based approaches that typically use VGG features for matching semantic information.

## 3 Proposed Method

In this section, we describe the proposed methodology in detail, including the network architecture design and the loss functions used.

### 3.1 Overview

The objective of our method is to improve single image reflection removal using semantic cues from easily sourceable reflection-free images. We develop our algorithm based on the following design principles: a) Given an input image, the algorithm should be able to quickly retrieve a small subset of semantically similar images from a large database of reflection-free images ; b) The algorithm should also be capable of extracting features from the reflection-corrupted input image and search for similar features from this subset of reflection-free images. Hence, this feature extractor should be *reflection-aware*, i.e, the features extracted from semantically

similar images should be similar regardless of the presence of reflections ; c) The algorithm should then use these *reflection-aware* features to aid removal of reflections from the input image.

An overview of the proposed method is shown in Fig. 2. For a given input image, a coarse semantic search is first performed on a reflection-free image database to obtain a set of $s$ images semantically similar to the input image. The database comprises of ~ 50,000 diverse images sourced from publicly available datasets [57] [60], that are predominantly reflection-free. The set of $s$ images are then passed through the proposed Reflection Aware Feature Extractor (RAFENet) to generate a Semantic Feature Dictionary (SFD). The SFD contains features that are mostly invariant to the presence of reflections. The SFD, along with the input image is then passed to the proposed SFDNet to generate the reflection-free output image. The salient components of the proposed pipeline is detailed in the following subsections.

### 3.2 Coarse Semantic Search

We propose a lightweight Coarse Semantic Search (CSS) module that retrieves a set of $s$ images similar to the input image from the database of reflection-free images in an efficient manner. We use a pre-trained VGG network as an image descriptor to encapsulate the semantic information of an image into a vector. Given an input image, we first extract the VGG image descriptors. Next we use cosine similarity metric to compare the input image descriptor with the descriptors pre-computed from the reflection-free images. The images with top 10 similarity scores are then passed to the proposed pre-trained *reflection-aware* feature extractor to compute a Semantic Feature Dictionary (SFD). An example of the frames obtained from the CSS module is shown in Fig. 2. It can be seen that the CSS module is able to find images with similar semantic or texture content as that of the provided input image. Please refer to the supplementary material for more examples of similar images from CSS module.

### 3.3 Reflection Aware Feature Extraction

While VGG based feature extractor can encapsulate the overall semantic content of an image, they are not designed to extract fine features from reflection-corrupted images consisting of superimposition of multiple image layers. The features extracted from the reflection-corrupted input image and reflection-free image database

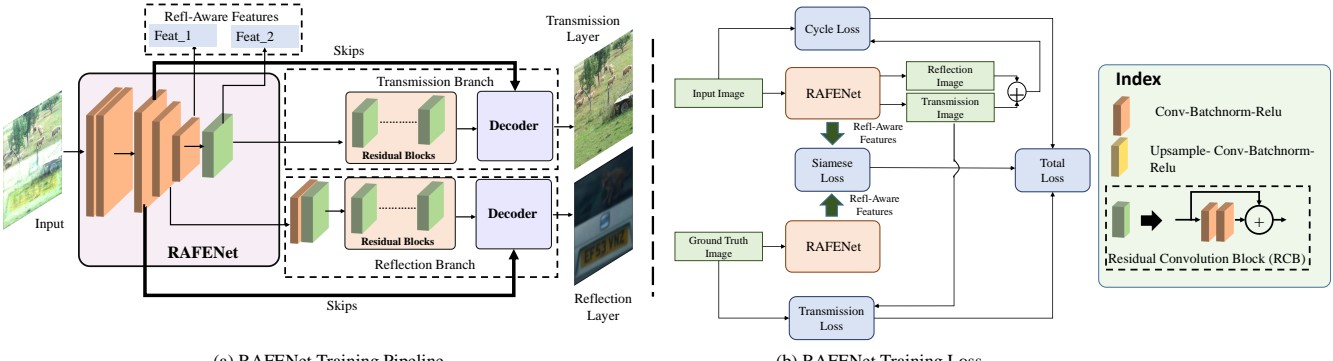

(a) RAFENet Training Pipeline

(b) RAFENet Training Loss

**Figure 3: (a) Architecture of the proposed RAFENet used to extract *Reflection Aware* features. The features marked $Feat_1$ and $Feat_2$ are used in subsequent stages. (b) Reflection Aware Loss Function consists of three components: Siamese loss, cycle loss and transmission loss. Input and ground truth images are passed to the RAFENet and the generated feature maps are used to compute Siamese loss. The transmission and reflection images are added together and compared with input image to compute cycle loss. The transmission image is compared with ground truth for transmission loss.**

should be *reflection-aware*; i.e, similar features should be extracted from semantically similar images regardless of the presence of the reflections. This is essential to effectively search for features related to the transmission content from the semantic feature dictionary. We propose a novel reflection-aware feature extractor (RAFENet) to generate meaningful reflection aware features from the input image as well as from reflection-free images.

*3.3.1 RAFENet Architecture.* The proposed RAFENet consists of an encoder block and a Residual Convolution Block (RCB). As shown in Fig. 3 (a), the encoder block consists of 2 conv-batchnorm-relu blocks of stride 1 ($CBR_1$), followed by 3 conv-batchnorm-relu blocks of stride 2 ($CBR_2$). The output from the RCB block is passed to a transmission branch, while the output feature map from the penultimate $CBR_2$ is passed to a reflection branch. We use separate branches so that features corresponding to reflection and transmission layers can be localized to their respective branches. This allows us to use features from the transmission branch as the required reflection-aware features. Both the transmission and reflection branches consists of further RCBs followed by decoder blocks. The decoder consists of 3 upsample-conv-batchnorm-relu blocks, followed by another conv-batchnorm-relu block and a convolutional layer with linear activation function. We use upsampling instead of deconvolution layers to avoid checkerboard artefacts. We also provide skip connections between the encoder block and decoder blocks. The output feature maps marked as $Feat\_1$ and $Feat\_2$ in Fig 3 (a) are used as the reflection-aware features. These layers are chosen empirically so that the resultant receptive fields of the extracted features are of the appropriate size for effective semantic transfer between the images.

*3.3.2 RAFENet Training Loss.* We propose a novel loss function to train RAFENet considering the following objectives: a) The features extracted in the transmission branch should be reflection aware; b) The transmission branch and reflection branches should generate transmission and reflection components of the input image respectively. An overview of the proposed reflection aware loss function is shown in Fig. 3(b). We use 3 components in the proposed loss

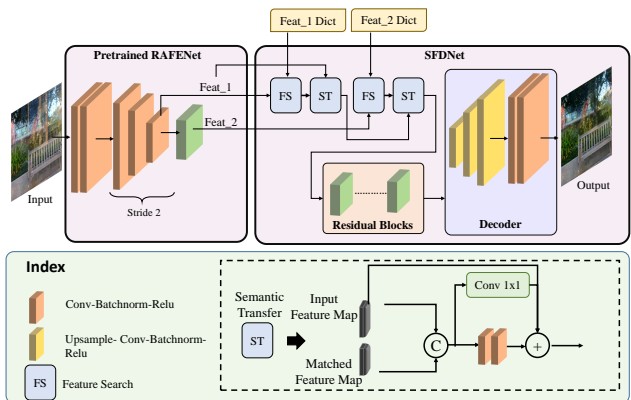

**Figure 4: Semantic Feature Dictionary Network (SFDNet). The input is first passed through the pre-trained RAFENet. Next, the Feature Search (FS) module searches for the most similar features from the SFD ($Feat_1$ and $Feat_2$ dicts) for semantic transfer.**

function: transmission loss, cycle loss and siamese loss. Each of these components are detailed in this section.

**Transmission Loss ($\mathcal{L}_{trans}$):** This loss function is applied on the output from the transmission branch so that features specific to transmission layer are learned. It is defined as:

$$\mathcal{L}_{trans} = \mathcal{L}_{I2I}(\mathcal{N}_t(I), I_g) \tag{2}$$

Here, $I$ denotes the input image, $I_g$ denotes the reflection-free ground truth image, and $\mathcal{N}_t(I)$ denotes the outputs from the transmission branch. $\mathcal{L}_{I2I}(I_1, I_2)$ is an Image-to-Image loss function [34] defined as follows:

$$\mathcal{L}_{I2I}(I_1, I_2) = 0.6\|I_1 - I_2\|_1 + 0.6\|I_1 - I_2\|_2 +$$
$$0.4\mathcal{L}_{grad}(I_1, I_2) + 0.8\mathcal{L}_{cl}(I_1, I_2) \tag{3}$$

where $\mathcal{L}_{grad}$ denotes $L_1$ in the gradient space and $\mathcal{L}_{cl}$ denotes contextual loss, which is a feature space similarity loss introduced by [30].

**Cycle Loss($\mathcal{L}_{cycle}$):** The sum of the outputs from the transmission and reflection branches should generate the original input image. This property can be used to implicitly constrain the output of the reflection branch using a cycle loss as follows:

$$\mathcal{L}_{cycle} = \mathcal{L}_{I2I}(\mathcal{N}_t(I) + \mathcal{N}_r(I), I) \tag{4}$$

where $\mathcal{N}_r$ represents the reflection branch.

**Siamese Loss ($\mathcal{L}_{siam}$):** The objective of this loss function is to constrain the training of RAFENet such that $feat_1$ and $feat_2$ are similar for both reflection-corrupted input image ($I$) and reflection-free ground-truth image ($I_g$). To compute $\mathcal{L}_{siam}$, we perform two forward passes per iteration of the training. In the first forward pass, we provide $I$ as the input to the network to generate $\mathcal{N}_t(I)$, $Feat\_1(I)$ and $Feat\_2(I)$. Here, $\mathcal{N}_t(I)$ represents the output from the transmission branch and $Feat\_1(I)$ and $Feat\_2(I)$ represent the feature maps extracted from $I$. In the second forward pass, we provide $I_g$ as input to the the network and generates $\mathcal{N}_t(I_g)$, $Feat\_1(I_g)$ and $Feat\_2(I_g)$. The Siamese loss is then computed as follows:

$$\mathcal{L}_{siam} = \mathcal{L}_{I2I}(\mathcal{N}_t(I), \mathcal{N}_t(I_g)) + \|f_{Feat\_1}(I) - f_{Feat\_1}(I_g)\|_1 \\ + \|f_{Feat\_2}(I) - f_{Feat\_2}(I_g)\|_1 \tag{5}$$

This loss function enforces the network to learn $feat_1$ and $feat_2$ in a manner invariant to the presence of reflections in the image.

The overall reflection invariant loss function is then defined as:

$$\mathcal{L} = \mathcal{L}_{trans} + 0.1\mathcal{L}_{cycle} + 0.15\mathcal{L}_{siam} \tag{6}$$

## 3.4 SFDNet

*3.4.1 Network Architecture.* We propose Semantic Feature Dictionary Network (SFDNet), a novel network that utilizes the Semantic Feature Dictionary generated by our proposed RAFENet to enhance single image reflection removal. An overview of SFDNet is shown in Fig. 4. The input image is first passed through the pre-trained RAFENet to generate *reflection aware* features. These features are then passed through a Feature Search (FS) module which returns the most similar matching features from the semantic feature dictionary. Next, we use a Semantic Transfer (ST) module [60] to fuse the matched features from the FS module with the reflection-aware features extracted from the input image. The resultant feature maps are then passed through Residual Convolutional Blocks (RCB), each consisting of two Conv-Batchnorm-Relu layers with residual connections. Finally, a decoder consisting of 3 upsample blocks and 2 convolutional blocks are used to generate the final output. We also provide skip connections between the feature extractor and decoder blocks. More details on FS and ST modules are discussed in this section.

**Feature Search (FS):** Given a reflection aware feature extracted from the input image, the objective of FS module is to retrieve the most similar feature from the semantic feature dictionary. Trivially

searching in the SFD is highly inefficient resulting in huge computational overheads. To this end, we propose a novel module for FS using lightweight differentiable operations. The proposed FS module can be plugged into any network architecture for efficient feature search. Let $f(I)$ be the feature map of size ($H \times W \times C$) extracted from the input image. We represent this feature map using the 2D matrix $\mathcal{F}$ consisting of $H.W$ rows and $C$ columns. Each row $i$ in this matrix ($\mathcal{F}[i]$) represents a $C$ dimensional feature vector extracted from the input image. We also define a matrix $\mathcal{F}_{sfd}$ of dimensions ($s.H.W \times C$) to represent the features in the SFD. Here, $s$ denotes the number of similar images obtained using coarse semantic search. For each feature in $\mathcal{F}$, the FS module searches for the most similar feature in $\mathcal{F}_{sfd}$ using cosine similarity metric and generates a matched feature map ($\mathcal{F}_m$). We use matrix operations for efficient computations. First, we construct a cosine similarity matrix as follows:

$$\mathcal{F}_{cos} = \frac{\mathcal{F}\mathcal{F}_{sfd}^T}{\mathbf{F}\mathbf{F}_{sfd}^T} \tag{7}$$

where $\mathbf{F}$ and $\mathbf{F}_{sfd}$ are norm vectors defined as follows:

$$\mathbf{F}[i] = \|\mathcal{F}[i]\|_2 \tag{8}$$

Here $\mathbf{F}$ and $\mathbf{F}_{sfd}$ are of dimensions ($H.W \times 1$) and ($s.H.W \times 1$) respectively and $\mathcal{F}_{cos}$ is a matrix of dimensionality ($H.W \times s.H.W$). Every element $\mathcal{F}_{cos}[i][j]$ in the cosine similarity matrix represents the cosine similarity of the $i^{th}$ input feature map with the $j^{th}$ feature map from the SFD. Next, an index matrix representing the indices of the most similar features in the SFD is constructed as follows:

$$\mathcal{F}_{ind}[i] = \arg\max F_{cos}[i] \tag{9}$$

Here, $\mathcal{F}_{ind}$ is a matrix of dimensions ($H.W \times 1$). Each element $\mathcal{F}_{ind}[i]$ denotes the index of the most similar feature from $\mathcal{F}_{sfd}$. The matched feature is then constructed as follows:

$$\mathcal{F}_m[i] = \mathcal{F}_{sfd}[\mathcal{F}_{ind}[i]] \tag{10}$$

$\mathcal{F}_m$ is a matrix of dimensions ($H.W \times C$) consisting of the most similar reflection-aware feature for each input feature vector.

**Semantic Transfer (ST):** We use a Semantic Transfer block similar to the one proposed in [60] to transfer the features from $\mathcal{F}_m$ to the subsequent features. A schematic of the ST module is shown in Fig. 4. Specific implementation details of this module is provided in the supplementary material.

*3.4.2 SFDNet Loss Function.* We use image-to-image loss function defined in Eq. 3 to train the proposed SFDNet.

## 4 Large Scale Reflection Removal Dataset

The efficacy of deep learning based image restoration is dependent on the availability of a large dataset of paired images, consisting of both reflection-corrupted and ground truth images. However, existing paired datasets for reflection removal [17, 49] consists of $\sim 1000$ image pairs, which is not sufficient for generalization. Wan *et.al* [48] proposed a large scale Reflection Image Dataset (RID) consisting of 3250 images. However, this dataset consists only reflection layer images, and proposes to construct synthetic training dataset by combining the provided reflection layer images with

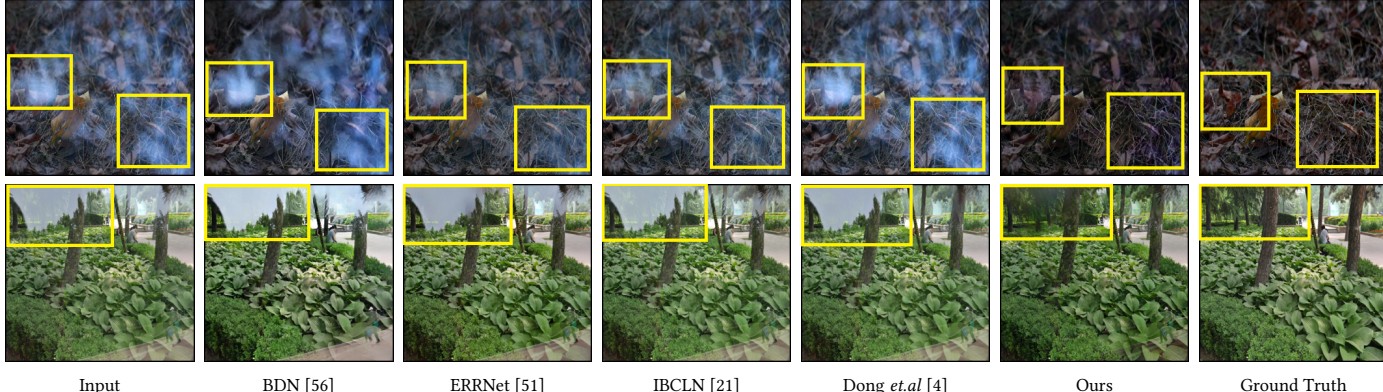

| Input | BDN [56] | ERRNet [51] | IBCLN [21] | Dong *et.al* [4] | Ours | Ground Truth |

**Figure 5: Comparison against state-of-the-art methods on Berkeley20 dataset (top) and ERR-DSLR50 dataset (bottom). Our method is able to remove reflections much better (Yellow boxes)**

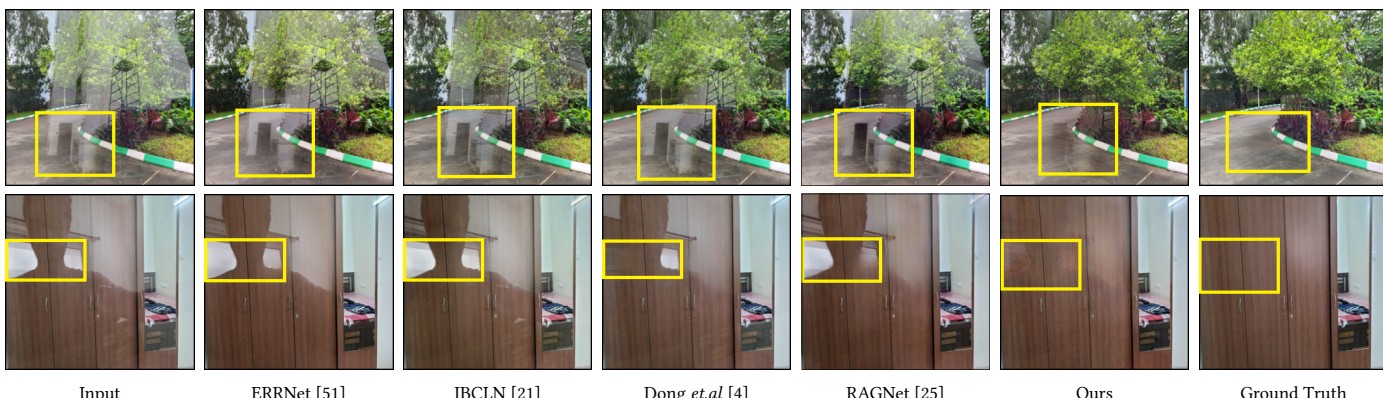

| Input | ERRNet [51] | IBCLN [21] | Dong *et.al* [4] | RAGNet [25] | Ours | Ground Truth |

**Figure 6: Comparison against state-of-the-art methods on our LSRR Dataset. Yellow boxes show that our method removes reflections better**

reflection-free images from public datasets. However the inaccuracies in the mathematical models used in synthetic data creation often translates to lack of adaptability of these methods to real-life scenarios.

To alleviate these issues, we propose a Large Scale Reflection Removal (LSRR) dataset for training and testing purposes. The proposed dataset consists of 2650 high resolution images with real reflections and their corresponding reflection-free ground truth images. The dataset was captured using a smartphone in various indoor and outdoor locations and in different lighting conditions. We use a portable glass panel to create images with and without reflections. The smartphone was affixed to a tripod so that the input image is aligned with the ground truth image. We also varied the angle of the portable glass with respect to the smartphone to simulate real-life capture conditions. Our dataset consists of images with strong as well as weak reflection components for better variability and generalization. To the best of our knowledge, the proposed dataset is the largest dataset consisting of images with real reflection and their corresponding aligned ground truths. We use 2550 image pairs for training and 100 images for benchmarking. Please refer to the supplementary material for some examples of image pairs from our dataset.

## 5 Experiments and Results

### 5.1 Datasets

*5.1.1 Training Dataset.* We use synthetic and real image data for training the proposed method. To create synthetic data for training, we make use of the datasets proposed in SRNTT [60] and PASCAL-VOC [5]. We use images from SRNTT dataset as the transmission layer, images from PASCAL-VOC as the reflection layer, and generate synthetic input images with reflections using the method proposed by [51]. We create a set of 11000 images pairs in this manner, of which, 1000 image sets are used for validation and the remaining for training. To train the proposed method using images with real reflections, we use the training set of the proposed LSRR Dataset consisting of 2550 image pairs.

*5.1.2 Testing Datasets.* We make use of the following datasets for evaluations: a) Berkeley-20 [59]; b) Postcard, Wildscene and Solid Object datasets from SIR2 [46]; c) Our LSRR test dataset; and d) ERR-DSLR50 [51] dataset. We use ERR-DSLR-50 dataset only for qualitative evaluations since this dataset consists of unaligned ground-truth images.

**Table 1: PSNR/SSIM comparisons against state-of-the-art on public datasets. Our method achieves the best average scores.**

| Dataset (size) | ERRNet-F [51] | IBCLN-F [21] | Dong *et.al* [4] | YTMT-F [9] | RAGNet [25] | SRNet [3] | Ours |
|---|---|---|---|---|---|---|---|
| SIR2-Postcard (199) | 20.39/0.829 | 23.87/0.876 | 23.72/**0.903** | 20.75/0.813 | 23.67/0.879 | 23.85/0.892 | **24.08**/0.881 |
| SIR2-Wild Scene (55) | 26.28/0.915 | 24.9/0.891 | 25.73/0.902 | 24.98/0.897 | 25.52/0.880 | 25.63/0.894 | **26.62/0.923** |
| SIR2-Solid Object (200) | 25.68/0.911 | 25.05/0.895 | 24.36/0.898 | 24.76/0.896 | 26.15/0.903 | 26.96/0.912 | **28.50/0.925** |
| Berkeley (20) | 22.52/0.803 | 20.95/0.760 | 23.31/0.812 | 20.83/0.753 | 22.95/0.793 | 23.58/0.803 | **24.59/0.818** |
| Average (474) | 23.39/0.872 | 24.36/0.880 | 24.20/0.8960 | 22.93/0.855 | 24.90/0.885 | 25.35/0.896 | **26.26/0.902** |

## 5.2 Implementation Details

We train the proposed RAFENet and SFDNet using PyTorch framework on a PC with NVIDIA Tesla V100 GPU and 32 GB RAM. Both RAFENet and SFDNet are trained till convergence using ADAM optimizer with an initial learning rate of $10^{-5}$ and a weight decay of 0.025. Please refer to the supplementary material for more details on training and implementation.

## 5.3 Comparison against state-of-the-art

We compare the proposed method against state-of-the-art methods in reflection removal, BDN [56], ERRNet [51], IBCLN [21], Dong *et.al* [4], YTMT [9], RAGNet [25] and SRNet [3]. For a fair comparison, we fine tune the methods with publicly available training codes (ERRNet [51], IBCLN [21] and YTMT [9]) using our proposed LSRR dataset. For the other methods we use the pre-trained models and inference codes provided by the authors for generating results.

*5.3.1 Qualitative Results.* We provide image comparisons against the aforementioned methods on public and our LSRR datasets in figures 5 and 6 respectively. We have shown results on challenging image sets consisting of strong reflections spread over a large area. It can be seen that the state-of-the-art methods fail to remove these reflections completely. Our method makes use of semantic information from the automatically extracted semantic feature dictionary to effectively remove reflections in these images. Hence, it can be seen that our method is able to remove reflections better than the state-of-the-art methods. Please refer to the supplementary material for more results.

*5.3.2 Quantitative Results.* We provide quantitative comparisons of our method against several state-of-the-art methods in reflection removal on multiple public datasets in Table 1 using PSNR and SSIM metrics. We use the notation -F to denote the methods fine tuned using our dataset. The proposed method is able to achieve better PSNR scores compared to the state-of-the-art methods on all the datasets. Our method also achieves the best SSIM scores on most of the datasets. Further, our method also achieves the best PSNR and SSIM scores averaged over all the public datasets.

We also compare against a self-semantic guided method by Liu *et.al* [26]. This method proposes to extract a segmentation map from the input images, and then use it to guide the reflection removal. However, unlike our method, this method does not utilize external semantic information, resulting in an inferior performance. This method generated output images with a PSNR of 23.75 and an SSIM of 0.8895 averaged over Berkeley and SIR2 datasets. On the other hand, our method outperform [26] on both these metrics obtaining scores of 26.26 and 0.902 respectively, resulting in an improved PSNR of ∼ 2.5 dB.

**Table 2: Comparison on the proposed LSRR Dataset (PSNR/SSIM). Original and fine tuned models are denoted by -O and -F respectively. Our method achieves the best PSNR/SSIM scores on public as well as our LSRR dataset.**

| Method | Public Datasets | Our Dataset (LSRR) | Average |
|---|---|---|---|
| ERRNet-O [51] | 23.41/0.869 | 20.68/0.804 | 22.91/0.857 |
| ERRNet-F [51] | 23.39/0.872 | 23.82/0.861 | 23.46/0.870 |
| IBCLN-O [21] | 24.23/0.870 | 20.57/0.807 | 23.57/0.858 |
| IBCLN-F [21] | 24.36/0.880 | 21.46/0.826 | 23.83/0.870 |
| Ours | **26.26/0.902** | **25.18/0.870** | **26.06/0.896** |

We also provide quantitative comparisons on the testing dataset of our proposed LSRR dataset in Table. 2. We compare our method against original (-O) and fine-tuned (-F) models of ERRNet and IBCLN methods. It can be seen that our method outperforms the existing methods on both PSNR and SSIM metrics on both public as well as our LSRR datasets. It can also be observed that the methods fine tuned using our training dataset (-F) significantly outperforms the original models (-O) on LSRR testing dataset. Moreover the fine tuned models are able to obtain comparable or better scores compared to original models on public datasets. This show the efficacy of the proposed LSRR dataset, in advancing further research in reflection removal.

## 5.4 Ablation Studies

In this section, we provide ablation studies to justify the importance of the novel components introduced in our methodology. All the experiments are evaluated on SIR2-Wildscene dataset. We also provide more experimental studies in supplementary material, including details on network complexity, impact of SFD size, extension towards an interactive approach and a detailed analysis of reflection-awareness.

**Impact of Image Database:** First, we analyze the impact of using image database of reflection-free images on the final output image. For this experiment, we evaluate the network without providing the SFD extracted from an external database of images. The result of this experiment is provided in row 1 of Table 3. It can be seen that this experiment results in a PSNR drop of ~1.3 dB compared to our proposed method which used image database (row 4 of Table 3). We also provide qualitative image comparisons for this experiment in Fig. 7 (top). It can be seen that the proposed method is able to remove reflections much better with the help of semantic content from external image database. This is evident especially in images with strong reflection components.

**Impact of Semantic Transfer:** Next, we evaluate the impact of Semantic Transfer module (ST) by training a network without ST

**Table 3: Ablation studies evaluated on Wildscene dataset. Image Database, Semantic Transfer module and RAFENet features improve the performance of the method.**

| Sl No. | Image Database | Semantic Transfer | Feature Extractor | PSNR | SSIM |
|---|---|---|---|---|---|
| 1 | × | × | RAFENet | 25.29 | 0.8897 |
| 2 | √ | × | RAFENet | 25.83 | 0.8949 |
| 3 | √ | √ | VGG | 25.95 | 0.8927 |
| Ours | √ | √ | RAFENet | **26.62** | **0.9232** |

**Table 4: Impact of various components of the proposed reflection aware loss function, evaluated on Wildscene dataset. Our proposed loss function achieves the best PSNR/SSIM scores.**

| Loss Function | PSNR (dB) | SSIM |
|---|---|---|
| only $\mathcal{L}_{trans}$ | 26.2 | 0..9013 |
| $\mathcal{L}_{trans} + \mathcal{L}_{cycle}$ | 26.25 | 0.9034 |
| $\mathcal{L}_{trans} + \mathcal{L}_{cycle} + \mathcal{L}_{siam}$ (Ours) | **26.62** | **0.9232** |

module. In this experiment, the matched features from the FS module are directly appended to the features extracted from the input image, without passing them thought the ST module. These results are summarized in row 2 of Table 3. It can be seen that the model without ST module resulted in a PSNR drop of ∼ 0.8 dB.

**Impact of Reflection Aware Features:** We also provide an ablation study to analyze the impact of the proposed Reflection Aware Feature Extractor (RAFE) on the output image quality. For this experiment, we used a pre-trained VGG as a feature extractor to extract features from the input image. We used the layers *conv2_2*, *conv3_4* and *conv4_2* of the VGG network for feature extraction. The extracted features were then passed through Feature Search and Semantic Transfer modules and the remaining layers of the SFDNet. We trained this network using the same training settings as that used for the original SFDNet. These results are summarized in row 3 of Table 3. It can be seen that replacing the proposed RAFE feature extractor with VGG results in a PSNR degradation of ∼ 0.55 dB. We also provide qualitative results on SRNTT validation set in Fig. 7 (bottom). It can be seen that the results obtained from SFDNet using VGG feature extractor is unable to remove the reflections completely, resulting in an output image with lower visual quality. On the other hand, SFDNet trained using the proposed Reflection Invariant feature extractor is able to remove reflections much better. This experiment shows the importance of *Reflection Aware* Feature Extraction for effective transfer of related semantic content from the reflection-free database of images.

**Loss Component Analysis for RAFENet:** In this section, we analyze the impact of the various components used in the proposed Reflection Aware loss function. For this experiment, we train the RAFENet using two different configurations: a) Only transmission loss ($\mathcal{L}_{trans}$); and b) transmission loss and cycle loss ($\mathcal{L}_{cycle}$). The resultant feature extractor is then loaded into the SFDNet and trained for reflection removal. The final PSNR and SSIM scores of the outputs generated by these configurations are compared with that generated by the proposed method using all the three losses ($\mathcal{L}_{trans} + \mathcal{L}_{cycle} + \mathcal{L}_{siam}$) . These results are summarized in Table. 4. It can be seen that removing $\mathcal{L}_{siam}$ results in a PSNR degradation

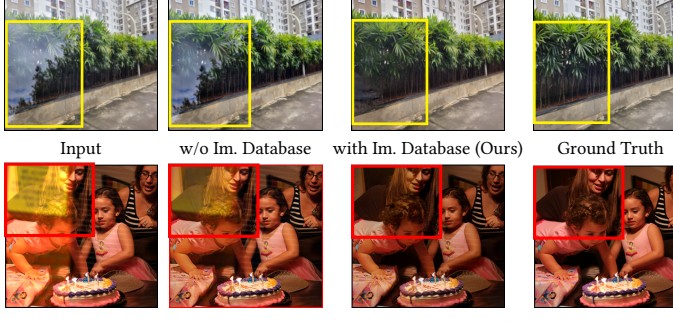

| Input | w/o Im. Database | with Im. Database (Ours) | Ground Truth |
|---|---|---|---|

| Input | VGG + SFDNet | RAFE + SFDNet (Ours) | Ground Truth |
|---|---|---|---|

**Figure 7: Ablation Studies. Top row: Impact of image database. Our method (Col 3) using image database is able to remove reflections much better; Bottom row: Impact of RAFENet. VGG based feature extractor (Col 2) results in inferior performance compared to our method using RAFENet (Col3).**

of ∼0.4 dB ( row 2 of Table. 4). Removal of $\mathcal{L}_{cycle}$ results in a further PSNR reduction of 0.05 dB. From this experiment, it can be inferred that $\mathcal{L}_{siam}$ contributes strongly to the model accuracy. This is expected, since removal of $\mathcal{L}_{siam}$ impact the ability of RAFENet to generate *reflection aware* features.

## 6 Conclusion

We present a novel methodology, termed $R^2SFD$, to improve reflection removal using semantic information extracted from easily available reflection free images. Our methodology consists of two novel network architectures, termed RAFENet and SFDNet. RAFENet is designed to extract *reflection aware* features that are mostly invariant to the presence of reflections. The SFDNet is designed to utilize the reflection-aware features extracted from the image database to improve removal of reflections from the input image. We also introduce a new dataset for reflection removal, termed Large Scale Reflection Removal dataset, consisting of 2650 images with real reflections and their corresponding ground truth images. To the best of our knowledge, the proposed dataset is the largest available dataset for reflection removal consisting of real image pairs. The proposed method achieves superior results both qualitatively and quantitatively against state-of-the-art methods on public real datasets as well as our LSRR dataset. We also conduct extensive ablation studies to show the impact of the external image database, Semantic Transfer module and RAFENet on the final output image. We further analyze the impact of various components of the proposed loss Reflection Aware loss function. These experiments show that proposed novel components and the new dataset enable us to achieve state-of-the-art results in reflection removal

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
