# OpenReview forum: "R2SFD: Improving Single Image Reflection Removal using Semantic Feature Dictionary"
_acmmm.org/ACMMM/2024/Conference — MM2024 Poster_

### Official Review · Reviewer_GHED · 2024-05-22

**Rating:** 4
**Confidence:** 3

**Summary:**

The R2SFD introduces an innovative approach to reflection removal by effectively leveraging a semantic feature dictionary (SFD) derived from reflection-free images. This represents a significant advancement over traditional methods that mainly rely on cues from corrupted input images. The paper also contributes a new dataset, Large Scale Reflection Removal (LSRR), which consists of 2650 image pairs that could be immensely beneficial for training and benchmarking in future studies.

**Strengths:**

1. The integration of RAFENet and SFDNet to extract and utilize reflection-invariant features is both novel and technically sound.
2. The method outperforms state-of-the-art techniques in qualitative and quantitative evaluations across multiple public datasets and the newly introduced LSRR dataset.

**Limitations:**

1. The method’s performance heavily relies on the availability of a large, semantically rich external database. How should the absence of certain semantic content in the dataset be addressed? Will this omission impact the effectiveness of the model?
2. How can the effectiveness of the reflection-free image database be ensured?
3. The paper lacks a comparative analysis of the computational complexity of the proposed method against existing methods.
4. The authors should consider incorporating a greater proportion of recent studies, particularly those published in 2023 and 2024, to ensure the paper's relevance and comprehensiveness.

**Suitability:**

2

---

### Official Review · Reviewer_deJy · 2024-05-23

**Rating:** 4
**Confidence:** 3

**Summary:**

The paper presents a novel approach to the challenging problem of single image reflection removal. The proposed method uses a Semantic Feature Dictionary (SFD) constructed from a database of reflection-free images. To further enhance reflection removal, the authors introduce the Large Scale Reflection Removal (LSRR) dataset, consisting of 2650 image pairs from various real-world reflection scenarios. The method shows superior results both qualitatively and quantitatively compared to existing techniques.

**Strengths:**

1. The introduction of a Semantic Feature Dictionary (SFD) and the innovative use of a Reflection Aware Feature Extractor (RAFENet) present a novel and promising approach to the problem of single image reflection removal.
2. The LSRR dataset  providing a diverse and representative set of reflection scenarios.
3. The paper addresses a practical problem with significant real-world applications, and the proposed method shows promise for improving the quality of images in various scenarios.

**Limitations:**

1. The paper proposes a novel reflection removal method using a SFD and a Reflection Aware Feature Extractor (RAFENet). While the approach appears promising, some aspects lack detailed explanation. For instance, how does RAFENet ensure that the extracted features are invariant to reflections? Additionally, visualizations such as feature maps could be included to demonstrate the model's performance in handling reflections.
2. While the paper introduces a Reflection Aware Feature Extractor (RAFENet) as a key component of the proposed method, it does not provide specific details about the data format or representation of these Reflection Aware Features. This lack of information makes it difficult to understand how these features are utilized within the network and how they contribute to the reflection removal process.
3. The paper introduces a new Large-Scale Reflection Removal (LSRR) dataset, but it lacks detailed descriptions regarding the diversity and representativeness of this dataset. While it is mentioned that the dataset contains 2650 image pairs, there is insufficient information about the sources, types of scenes, and diversity of reflection scenarios.The authors should provide more comprehensive information about the LSRR dataset, such as the sources of the image pairs, the distribution of scenes, the intensity, and types of reflections. This would better demonstrate the diversity and representativeness of the dataset, enhancing its credibility and application potential.
4. The paper compares the proposed method with several existing reflection removal techniques. However, it appears that some of the compared methods on the proposed LSRR dataset are not the most recent or advanced in the field. The most recent research work compared seems to be for 2020? The authors should update their comparison to include more recent and advanced reflection removal methods published in the last couple of years. This will provide a more accurate and up-to-date evaluation of the proposed method's performance.

**Suitability:**

2

---

### Official Review · Reviewer_W5i4 · 2024-05-23

**Rating:** 4
**Confidence:** 3

**Summary:**

The study presents "R2SFD," an innovative framework for effectively removing reflections from single images. Central to this approach is the utilization of a semantic dictionary, derived from a database of clean images, to assist in the accurate discrimination and subtraction of reflective components. The study also introduces a dataset designated for reflection removal tasks. The proposed method demonstrates notable improvements over existing models, showcasing its robustness through extensive tests on a variety of datasets.

**Strengths:**

1. The deployment of a Semantic Feature Dictionary (SFD) presents a novel strategy for addressing the issue of reflections in images. By relying on semantic features extracted from a clean image database, the method introduces an innovative pathway to reflection removal that goes beyond conventional techniques.
2. The creation and utilization of the Large Scale Reflection Removal (LSRR) dataset represents a contribution to the research community.
3. The proposed R2SFD framework has demonstrated superior performance in comparison to existing techniques.

**Limitations:**

1. The methods section is not clear enough, for example, line 403 “These layers are chosen empirically”, but there is no further explanation and no reference to related work.
2. The proposed method relies on a Semantic Feature Dictionary (SFD) built from a database of clean, reflection-free images. This dependence suggests that performance may be limited in cases where such clean images are not readily available or where the reflection properties differ significantly from those in the database.
3. The inclusion of multiple components like RAFENet and the SFDNet might increase the complexity of the model. Dealing with reflections in real-time scenarios may pose a challenge due to computational demands.

**Suitability:**

2

---

### Official Review · Reviewer_y7w1 · 2024-06-03

**Rating:** 4
**Confidence:** 4

**Summary:**

The paper introduces a novel method for single image reflection removal by constructing a semantic feature dictionary.  This dictionary is created using a reflection-aware feature extractor that captures features that remain consistent across images with and without reflections. To further enhance its effectiveness, this paper also introduces a new Large Scale Reflection Removal (LSRR) dataset.  The proposed method demonstrates superior performance in both visual quality and quantitative metrics when compared to state-of-the-art methods on public real-world datasets as well as the LSRR dataset.

**Strengths:**

1. The proposed method introduces an innovative idea of using external reflection-free databases to enhance the reflection removal process, which could potentially lead to more accurate and effective results.
2. The introduction of the LSRR dataset is commendable as it provides a large-scale, high-quality benchmark for researchers to evaluate their methods, promoting the development of the field.
3. The proposed method outperforms state-of-the-art methods in reflection removal both qualitatively and quantitatively.

**Limitations:**

1. There are some errors in this paper. For example, the SSIM of 'only Ltrans' has two decimal points in Table 4.
2. This paper does not compare the model complexity with other methods, but only compares the different components of its own model.
3. The testing time is not compared.
4. While the use of external reflection-free semantic content databases is innovative, it may be constrained by the size and diversity of available databases. Smaller or niche databases might not provide sufficient information to comprehensively improve reflection removal in single images. Additionally, the cost of acquiring and maintaining these databases could be substantial.
5. The LSRR (Large Scale Reflection Removal) dataset may suffer from data imbalance due to the variety of reflection conditions in real-world scenarios. If the training set only includes certain types of reflections, the model might perform poorly on unseen types. Addressing data imbalance is crucial for improving the model’s generalization capabilities.
6. The effectiveness of the proposed method is highly dependent on the quality of the external databases used. If the databases contain errors or inconsistencies, it could negatively impact the performance of the reflection removal algorithm.

**Suitability:**

2

---

### Meta-Review · Area_Chair_3JkE · 2024-07-01

**Recommendation:** Accept (Poster)
**Confidence:** 5

**Metareview:**

Reviewers acknowledged the good results and novel use of an external database. The initial scores are all positive and final scores are  mixed, with both lowering scores and increasing scores happening. The main concerns from all reviewers are also from its novel design, that is the external database. While it shows effect on the good side, there are concerns with it. The reviewers questioned if it will also hurt in some cases, if the input data is outside the LSRR distribution, or if the semantic search fails, etc. The authors partially answered these questions but not all reviewers are fully satisfied, and there are lowering scores happening. After careful discussion and consideration, the ACs decided to accept this paper to encourage this kind of attempt.

However, the ACs share the same concern with the reviewers about the potential negative side of using an external database. The authors are required to consider all the concerns raised by the reviewers and reveal the limitations by showcasing what the results will be if some part is not working properly as expected, or to what extent the method is robust. Revealing limitations is also part of research and will benefit the readers of this paper in this direction.